# Bacteria of Zoonotic Interest Identified on Edible Freshwater Fish Imported to Australia

**DOI:** 10.3390/foods12061288

**Published:** 2023-03-17

**Authors:** Michelle Williams, Shokoofeh Shamsi, Thomas Williams, Marta Hernandez-Jover

**Affiliations:** 1School of Agricultural, Environmental and Veterinary Sciences & Graham Centre for Agricultural Innovation, Charles Sturt University, Wagga Wagga, NSW 2650, Australia; 2Institute for Future Farming Systems, CQUniversity, Rockhampton, QLD 4701, Australia

**Keywords:** zoonotic bacteria, imported fish, Australia

## Abstract

Previous research has shown that freshwater edible fish imported into Australia are not compliant with Australian importation guidelines and as a result may be high risk for bacterial contamination. In the present study, the outer surface of imported freshwater fish were swabbed, cultured, confirmatory tests performed and antimicrobial patterns investigated. Channidae fish (Sp. A/n = 66) were contaminated with zoonotic *Salmonella* sp./*Staphylococcus aureus* (n = 1/66) and other bacteria implicated in cases of opportunistic human infection, these being *Pseudomonas* sp. (including *P*. *mendocina* and *P. pseudoalcaligenes* (n = 34/66)); *Micrococcus* sp. (n = 32/66); *Comamonas testosteroni* (n = 27/66) and *Rhizobium radiobacter* (n = 3/66). Pangasiidae fish (Species B/n = 47) were contaminated with zoonotic *Vibrio fluvialis* (n = 10/47); *Salmonella* sp. (n = 6/47) and environmental bacteria *Micrococcus* sp. (n = 3/47). One sample was resistant to all antimicrobials tested and is considered to be Methicillin Resistant *S. aureus*. Mud, natural diet, or vegetation identified in Sp. A fish/or packaging were significantly associated with the presence of *Pseudomonas* spp. The study also showed that visibly clean fish (Sp. B) may harbour zoonotic bacteria and that certain types of bacteria are common to fish groups, preparations, and contaminants. Further investigations are required to support the development of appropriate food safety recommendations in Australia.

## 1. Introduction

The growth of the fish farming (FF) sector in many countries, the ‘blue revolution’ [1], occurred without regulatory guidance or enforcement of environmental or food safety standards [2]. High stocking density/fish stress [3,4,5], lack of appropriate biosecurity barrier controls [6] and inadequate isolation of seed stock [7,8,9] have all contributed to the spread of bacteria in FF. As a consequence, the use of prophylactic and therapeutic antibiotics has increased to compensate for inadequate sanitary practices [10,11,12,13,14,15]. Approximately 90% of global FF takes place in countries that have no or few effective regulations for antibiotic use [16,17,18,19,20]. Despite the advances in bacterial control in FF, phytobiotics to improve disease tolerance [21,22], phytocompounds with antimicrobial activity [23], vaccines and improved husbandry practices [24,25,26], phage therapy and research into antimicrobial peptides [27] and biosynthesized nanoparticles [28,29], antibiotics are still injudiciously applied to control bacterial disease in some countries [30,31]. Antibiotics used to inactivate or inhibit bacteria may accumulate in the ponds, sediments and fish raised in the FF system [2]. A 2016 study showed results which supported the hypothesis that selection pressures present in FF favour rapid alterations in pathogenic bacterial populations. These alterations are likely to have enduring evolutionary effects on pathogen virulence [32]. According to Shi, et al. [33], bacterial pathogens are now the ultimate disease affecting aquatic species. This conclusion is supported by Cai, et al. [34], Hoai, et al. [35] and Reverter, et al. [36].

Many of the world’s largest producers of farmed fish use the VAC system (**V**uon: garden, **A**o: pond and **C**huong: livestock pen), fish systems for commercially intensive production [37], pond–dyke [38,39] and paddy-fish farming [40], lake/reservoir [41] and polyculture systems [42]. Fish may be fed food waste and animal manure [37], pelleted feed or raw feed, soybean and grass often grown in pond sludge [42], homemade diets consisting of food waste, such as chicken bones/viscera, and kitchen remains [43]. According to Chávez-Crooker and Obreque-Contreras [44], sediments organically enriched through intensive FF will produce an ecosystem which is dominated predominantly by bacteria. In addition, the extractive species, such as bivalves, aquatic plants and algae, which utilise water nutrients also accumulate antibiotics administered to other species in these intensive production systems [45]. Aquatic pollution discharged from households, farmed livestock, hospitals and untreated sewerage are also potential sources of contamination of resistant genes into water and sediment in FF facilities [46]. Preena et al. [47] notes that antibiotic resistance patterns can be transferred between clinically important strains through horizontal gene transfer, and that the majority of farmed fishes are contaminated with a broad spectrum of multiply resistant bacteria. In Bangladesh, a pattern of resistance was observed for *Klebsiella pneumoniae* and to a lesser extent *E. coli* and *P. aeruginosa* isolated from urinary tracts of infected women in Sylhet [48] and other recently isolated clinical samples in Jessore [49]. High patterns of resistance were correspondingly shown in *K. pneumoniae* and *E. coli* isolates from farmed Hilsa shad (*Tenualosa ilisha*) in the Dhaka and Chittagong regions [46] against ten antimicrobials [46,49]. According to Slettemeås, et al. [50], the global trade in animal products and food may represent significant dissemination routes which enable important resistant bacterial types to be introduced into geographic areas outside usual areas of endemicity.

In recent years many environmental bacteria previously considered of no consequence to human health have emerged as serious human pathogens. Certainly, ‘high risk environments’ for bacterial genetic resistance exchange are sewage, wastewater, discharge from hospitals, aquaculture and waste from livestock, agriculture and abattoirs [51,52,53,54], all of which are present in many of the main producers of freshwater fishery products (FP) exported to Australia.

A variety of zoonotic bacterial species have been identified in imported vacuum packed and frozen catfish (*Pangasius*) fillets [55,56,57]. *Vibrio fluvialis* and *Salmonella* spp., which cause intense diarrhoea, have been identified on fresh and frozen fish offered for retail sale [58,59,60] or in water or fish at FF facilities [61,62,63].

Zoonotic bacteria contaminating FP are able to infect humans in three ways: contact with the aquatic animal, FP or water that they inhabit [64], via consumption of under cooked FP [65], or from the toxin produced by the bacteria contaminating the FP [66]. Opportunistic infection of humans with non-zoonotic bacteria has also been reported following contact with contaminated environmental sources [67,68,69,70,71,72,73]. Zoonotic bacteria may not only cause illness if consumed, but according to Economou and Gousia [74], due to the over/misuse of antimicrobials in food production a reservoir of antimicrobial resistant bacteria can be transmitted to humans via contact, both direct or indirect. Bortolaia, et al. [75] hypothesises that humans may acquire antimicrobial resistant bacteria after consumption or handling of contaminated food. Therefore, in order to protect consumers and vulnerable members of the community, it is expected that testing procedures for detection of zoonotic bacteria in imported farmed fish should be robust.

As per the ‘*Imported Food Control Regulations 2019*’ [76] and the ‘*Imported Food Control Act 1992*’ [77], the relevant Australian minister may classify a certain imported edible product as ‘risk’ food and this is based on the product in question being a high or medium risk to public health. In Australia at the present time (2022), the only additional test applied to ‘risk’ imported fish, cooked, smoked, smoke flavoured or vacuum packed [78], is for the zoonotic bacterium *Listeria monocytogenes* (Figure 1) as per Schedule 27 ‘Microbiological limits in food’ in the Australia and New Zealand Food Standards Code [79].

Any additional tests applied to imported seafood must be justified according to the global trade agreements signed by Australia [80]. Williams et al. [81] developed a risk scoring system to identify countries as high risk for supply chain breaches which may impact the safety of edible finfish imported into Australia and to justify and prioritise examination of these products for supply chain breaches, such as zoonotic parasites/bacteria and physical and other biological hazards. Six predictor variables were scored to achieve an outcome variable of ‘*Freshwater fish high risk*’. Each country was allocated a unique numerical and anonymous identifier. Imported freshwater fish were examined from two of the highest scoring countries for supply chain breaches identified as Country 20 and 22 in Williams et al. [81] and in the present study. The Williams et al. [81] study identified a number of supply chain breaches, such as mud, vegetation, natural diet, snails and other physical hazards from poorly processed ‘consumer ready’ fish and in the same study the outer surface of each fish was swabbed for bacterial culture in the present study.

The aims of the current study were to identify and compare selected bacterial flora of imported edible freshwater finfish, Sp. A/Country 22 and Sp. B/Country 20, and describe the antimicrobial resistance pattern for identified bacteria. In addition, associations between the presence of bacteria and contaminants, such as mud, vegetation, and natural diet, were also explored.

The purpose of this study was to identify bacteria on edible fish imported into Australia, not to disadvantage an exporting country. Therefore, information that may lead to the identification of an included country, such as fish species names, has been omitted from the manuscript and auxiliary tables. Any published literature cited in text omitted to maintain country confidentiality is indicated with (*).

## 2. Materials and Methods

### 2.1. Sampling

A total of 60 entire imported Sp. A fish, 41 fillets of imported Sp. B fish and thawed ice samples ‘bag water’ from each of the 12 separate bags of fish were swabbed for bacterial culture using a sterile culture swab as per Saito et al. [82]. ‘Bag water’ and Sp. A and Sp. B results were combined to reflect 66 and 47 samples, respectively (Sp. A: 6 individual bags of fish (n = 60 fish) + 6 ‘bag water’ samples = 66 samples in total, and Sp. B (6 individual bags of fish (n = 41 fish) + 6 ‘bag water’ samples = 47 samples in total)). Both Sp. A and B were pre-packaged, frozen, obtained from a retail outlet and maintained frozen until thawed for sampling. Sample size was based on availability at the time fish were purchased. Species A had been frozen in a single layer (original packaging), in two separate rows, and were embedded in a solid block of ice in each of the 6 bags. Species B fish were frozen individually in each bag and each fillet was glazed with a thin covering of ice. A detailed description of the type and condition of fish packaging is included in Table 1. All fish were thawed, according to packet instructions, in a refrigerated unit (~24 hours (hrs)) in original unopened bags. The bag was split using sterile scissors without touching the fish or bag water. The skins of each fish, which were entire, were swabbed by gently scraping twice along the external surface of the lateral flank using the same sterile swab. For each fish fillet the swab was gently scraped twice along the outer lateral surface. The ‘bag water’ was sampled by dipping one sterile culture swab into the fluid once without touching the bag. Following sampling, the contents of each bag were examined and irregularities, such as the presence of mud, vegetation, food items and other debris, recorded for future statistical analysis. 

### 2.2. Bacterial Species Selection

Due to serious time constraints, it was not possible to culture broadly for all bacteria contaminating the outer surface of Sp. A and Sp. B. Specific bacteria targeted in this investigation were chosen based on the country of origin of the FP, the supply chain breaches, such as mud identified in Williams, et al. [81], water and sanitation information in Wendling, et al. [83], and published literature which describes each bacterium having been identified in FF, sediment/mud, water or FP in Countries 20 and 22 (*). Due to the identification of mud and other debris contaminating FP from Country 22 [81] and the low scores in Wendling, et al. [83] for water and sanitation in both countries, *E. coli* and *Pseudomonas* sp. were chosen as total faecal and water indicators as described in Odonkor and Ampofo [84], Rodrigues, et al. [85] and Odjadjare and Ebowemen [86]. *Salmonella* sp./*Staphylococcus aureus* were selected as an indicator of faecal/bacterial contamination during processing [87]. *Clostridium* sp. were selected as an indicator of excreta contamination from human sewerage/non-herbivorous wildlife as demonstrated in Vierheilig et al. [88] or as an indigenous bacterium of fish, from cross contamination during processing [87]. All targeted bacterial species in this study have been identified in either FF or FP in Countries 20 and 22 (*).

### 2.3. Diluent Preparation, Isolation, Enumeration and Identification

ThermoFisher selective media plates were used for the isolation of target bacteria. Brilliance *E. coli*/coliform (EC) agar (product code (PC): PP2609) was used to isolate *E. coli*, Baird Parker agar ((BP) PC: PP2069) was used to isolate *S. aureus*, TSC No Egg Yolk ((TSC) PC: PP2595) for *Clostridium* spp., Brilliance *Salmonella* Chromogenic media ((SAL) PC: PP2351) for *Salmonella* spp. and for *Pseudomonas* spp. CFC *Pseudomonas* selective agar was used ((CFC) PC: PP2161). Peptone water (PC: TM0057) (1.5 mL) was used as a diluent according to manufacturer instructions to inoculate TSC, BP, CFC plates. For SAL plates, the fish swab was added to 1.5 mL of buffered peptone (TM1558) in sterile Eppendorf^®^ tubes and incubated at 40 °C for 24–30 hrs according to product guidelines before inoculating each plate. All selective media plates were inoculated using a sterile 10 µL COPAN inoculating loop (PC: COP-H10) and then incubated at 35 °C for 48 hrs.

Bacterial colonies on plates were counted according to the quadrant method [89]. Bacterial colonies were marked with a highlighter on the culture plate as each colony was counted. As the antibiotics in selective agar can interfere with antimicrobial sensitivity and other confirmatory tests, bacterial isolates obtained from selective media plates (BP; TSC; SAL; CFC) and identified based on Gram stain were diluted in one mL of sterile saline and regrown on two separate Mueller Hinton agar plates ((MH) PCPP2096). One plate was used for antimicrobial disk diffusion susceptibility testing and the other for specific confirmatory tests. All sampling was conducted using a sterile 10 µL COPAN inoculating loop.

#### 2.3.1. Antimicrobial Resistance

Bacterial isolates were prepared according to the Kirby–Bauer Disk Diffusion Susceptibility Test Protocols [90]. Sterile saline was used for the bacterial suspension (variable dilutions) to achieve a 0.5 McFarland turbidity standard against a Wickerham card. The specific antimicrobial diffusion disks used were selected from the WHO [91] as critically important antimicrobials for human medicine. Four of the antimicrobials selected for evaluation were from categories C1 (‘*sole, or one of limited available therapies, to treat serious bacterial infections in people*’) and C2 (‘*used to treat infections caused by bacteria possibly transmitted from non-human sources, or with resistance genes from non-human sources*’) and one was in the C2 category only. All antimicrobials selected are used in FF or have been identified contaminating farmed freshwater fish in published literature [91,92,93,94,95,96,97,98,99,100]. Antimicrobial disks used in this study were obtained from ThermoFisher. Availability of some antimicrobial disks during COVID restricted a broader range of resistance testing. The following antimicrobial disks were used: (critically*/highly important#), Cefovecin* (3rd generation Cephalosporin) (30 µg PC: CT1929B), Cefoxitin# (2nd generation Cephalosporin) (30 µg PC: CT0119B), Ciprofloxacin* (Quinolone) (5 µg PC: CT0425B), Fosfomycin* (Phosphonic acid derivative) (50 µg PC: CT01838) and Vancomycin* (Glycopeptide) (30 µg PC: CT0058B) as per Okoh and Igbinosa [101], Tall, et al. [102], Kaysner, et al. [103] and Lesmana, et al. [104]. Only *Vibrio fluvialis* was tested for sensitivity to Colistin* (Polymyxin) (10 µg PC: CT0017B) and Cephalothin (1st generation Cephalosporin) (30 µg PC: CT0010B) which is not included in WHO [91] lists. Antimicrobial disks were manually placed on MH agar using sterile tissue forceps. Antimicrobial sensitivity evaluation was conducted 24 hrs following MH plate inoculation. Isolates from BP plates were cultured for a further 24 hrs as advised in the Clinical and Laboratory Standards Institute [105] (CLSI). Breakpoint zone measurements where possible were obtained from CLSI [105] and where a suitable category did not exist from Hudzicki [90], Reynolds [106] and Behera, et al. [107]. Breakpoint zones were described as either sensitive, intermediate, or resistant.

#### 2.3.2. Isolate Identification

Bacterial isolates from the second MH culture plate were Gram stained according to Beveridge [108] and were examined at 100X magnification with emersion oil using an Upright Motorized Microscope ECLIPSE Ni-E, Nikon, Japan, fitted with a computer screen and camera to observe bacteria morphology. All reagents were obtained from ThermoFisher. *Micrococcus* sp. and *Staphylococcus aureus* were differentiated using Coagulase (slide and tube) (PC: R21051), Oxidase (PC: MB0266B) and Catalase testing. *Salmonella* sp. was confirmed using the *Salmonella* Latex Test (DR1108A). Isolates from CFC *Pseudomonas* plates (15 tested) and Gram-negative oxidase positive isolates from TSC plates with the same growth morphology (12 tests), all pale-yellow small round/globular colonies (3 tests) and purple colonies from SAL plates were tested according to Rapid ID^TM^ NF Plus System (PC: R8311005). For Rapid ID™ testing 1 mL of Inoculation Fluid (PC: R8325102) was added to each bacterial isolate to achieve a visual turbidity of ≥1.0 and <3.0 according to McFarland standard. The bacterial solution was added to each Rapid ID™ rack and incubated for 4 hrs before scoring the ten wells as positive or negative according to the Rapid ID^TM^ NF Plus colour guide. Reagents Rapid NF Plus reagent (PC: R8311005), RapID™ Spot Indole Reagent (PC: R8309002) and RapID Nitrate A. Reagent (PC: R8309003) were then added to appropriate test wells according to product information. Wells 4–10 were rescored against the same colour guide. Results were entered into the Rapid NF Plus Code Compendium (ERIC^TM^) for identification and reports for each test downloaded. Following the 15 biochemical tests to confirm species identity of representative colonies grown on CFC *Pseudomonas* plates, other colonies were presumptively confirmed as *Pseudomonas* sp. based on growth on selective CFC media, negative Gram stain, characteristic morphology, positive Catalase and Oxidase, negative Coagulase and resistant to antimicrobials Cefovecin, Cefoxitin and Vancomycin. *Comamonas testosteroni* not tested using Rapid ID^TM^ NF Plus System (RNF Plus) were identified based on Gram stain, characteristic morphology, negative lactose-fermentation (MacConkey without Salt Agar Plates (PC: PP2016)), positive Catalase and Oxidase.

### 2.4. Data Analysis

Data were expressed as the actual number of fish contaminated and the range of bacterial colonies for each bacterial species or genera identified in Sp. A and Sp. B. The recovery rate (RR) of bacterial contamination in all fish was calculated according to Bush, et al. [109] and calculation of bacterial colonies based on Smith & Brown [110]. 

Associations between presence or absence of bacteria and a set of explanatory variables or potential factors were investigated using univariable logistic regression analyses, when sufficient data were available. Explanatory variables in the present study were based on the supply chain breaches identified in Williams, et al. [81] and included ‘mud’, ‘natural diet’, and ‘components of natural diet’ which were absent or present in each bag of fish. Analyses were conducted using statistical software ‘R’ (R Core Team 2020). The explanatory variables used were fish family, mud, vegetation, and natural diet. Best model fit was assessed using the Chi-Square Test and odds ratios and confidence intervals were calculated to quantify the strength of an association between the occurrence of bacteria and predictor. Only relationships that resolved to significance (*p <* 0.05) are reported in the results section. In some instances, insufficient data were available to develop a robust model. Either count data were one-sided or substantially unbalanced. These data were omitted from statistical analyses.

## 3. Results

The bacteria (cfu/g = colony forming units per gram) found in imported Sp. A and Sp. B are included in Table 2. Antimicrobial disks, Colistin (CT) and Cephalothin (KF), were tested as diagnostically significant for *V. fluvialis* only (Table 3). No growth of *E. coli* or *Clostridium* sp. was observed on EC or TSC indicator agars, respectively, and are omitted from results. 

### 3.1. Species A from Country 22

Mud was present in 46.9% of samples from Sp. A and represented by all fish in three of the six bags of fish examined. Natural diet (snail shells, cricket legs, other small crustaceans, and insect larvae) was identified in 65.1% of the samples, represented by four bags of Sp. A fish, and vegetation (green matter, sticks/twigs, and other decomposing vegetable matter) was identified in 18.1% of fish, all coming from one bag only of the Sp. A examined (Table 1).

*Pseudomonas* sp. was the dominant bacteria isolated from the skin of imported Sp. A fish with an overall RR of 51.5%, *Micrococcus* sp. had an RR of 48.5% and *S. aureus* (S700) an RR of 1.5% (Table 2). *Comamonas testosteroni* (syn. *Delftia testosteroni*) had an RR of 40.9%, and *Rhizobium radiobacter* was isolated from three samples (4.5%) (S468; S655; S690), with *Salmonella* sp. being isolated from one sample (1.5%) (S469).

#### 3.1.1. *Pseudomonas* sp.

Only *Pseudomonas* sp. was grown on CFC selective agar. Of the 66 swabs cultured, 34 were positive for *Pseudomonas* sp. with a mean cfu/g of 104.0. Of the 15 isolates tested using the RNF Plus, 60% (N = 9) were identified as *P. pseudoalcaligenes* and 40% (N = 6) as *P. mendocina* (Table 2). No difference in colony morphology was observed between *Pseudomonas pseudoalcaligenes* and *P. mendocina* on either CFC or MH agar at 24 hrs. Remaining samples were presumptively identified as *Pseudomonas* sp. based on tests in Table 3. Of the six *P. mendocina* identified using the RNF Plus there were no test contraindications and 100% of the samples tested returned a probability score of 92.11% and bio-score of 1/7 according to the ERIC^TM^ Compendium. Species level identification of *P. mendocina* was confirmed by glucose oxidation. Of the nine *P. pseudoalcaligenes* identified using the RNF Plus there were no test contraindications for eight of the samples tested and each returned a probability score of 97.21% and bio-score of 1/3. Sample S689 identified as *P. pseudoalcaligenes* had a probability score of 79.08%, a bio-score of 1/49 and slight overlap with *P. stutzeri.* Species level identification in all samples of *P. pseudoalcaligenes* was confirmed by glucose alkalisation.

A complete pattern of resistance was demonstrated by *Pseudomonas* sp. for Cefovecin, Cefoxitin and as expected to Vancomycin (Table 4).

#### 3.1.2. *Micrococcus* sp. and *Staphylococcus aureus*

Both *Micrococcus* sp. and *S. aureus* grew on BP agar. *Micrococcus* sp. was cultured in 32/66 samples and had a mean of 50.0 cfu/g and one sample only of *S. aureus* cultured with 27 cfu/g observed (Table 2). *Micrococcus* sp. and *S. aureus* were identified based on tests in Table 3.

#### 3.1.3. *Comamonas testosteroni* (syn. *Delftia testosteroni*)

Colonies on TSC were exclusively *C. testosteroni* cultured from 27/66 swabs and with a mean of 61.7 cfu/g (Table 2). Of the twelve *C. testosteroni* identified using the RNF Plus there were no test contraindications for 100% of the samples tested. Ten of the samples returned a probability score of 96.13% and bio-score of 1/9. Samples S439 and S649 returned a probability score of 95.45% and bio-score of 1/24 and showed a slight overlap with *Burkholderia cepacia.* Species level identification of *C. testosteroni* in all samples was confirmed by glucose alkalisation. *Comamonas testosteroni* was then presumptively identified based on negative Gram stain, characteristic morphology, negative lactose-fermentation, positive to Catalase and Oxidase (Table 3).

#### 3.1.4. *Rhizobium radiobacter*

*Rhizobium radiobacter* was cultured on MH agar from 3/66 swabs and had a mean of 7.6 cfu/g (Table 2). All *A. radiobacter* were identified using the RNF Plus and all samples tested returned a probability score of 99.9% and a bio-score of 1/8 (Table 3). All three isolates grew as red colonies on MacConkey agar as described in Chanza, et al. [111] from an isolate in a human pulmonary abscess case, Spain, identified using mass spectrometry MALDI-TOF^®^, Beckman Coulter^®^ biochemical profile and molecular method.

#### 3.1.5. *Salmonella* sp.

*Salmonella* sp. on SAL agar was cultured from one sample only and 32 cfu/g was observed (Table 2). Sample (S469) was tested using the *Salmonella* Latex Test and returned a positive result (Table 3).

### 3.2. Species B Fish from Country 20

*Vibrio fluvialis* was the dominant bacteria isolated from the outer surface of imported Sp. B fillets with an RR of 21.3%, followed by *Micrococcus* sp. with an RR of 6.0% and *Salmonella* sp. with an RR of 1.5%. One sample (S1013) was positive for *S. aureus*. *Vibrio fluvialis* was identified from four of the six bags tested, with bags one and four each having four positive samples. Within the ten positive samples of *V. fluvialis*, three were identified in the ‘bag water’ (S878; S908; S1007). *Micrococcus* sp. and *Salmonella* sp. were all recovered from the outer surface of the fillets from two and three bags, respectively (Table 2). Bag three had the most diverse bacterial flora with *V. fluvialis* (S908; S912), *Micrococcus* sp. (S911) and *Salmonella* sp. (S910; S912) identified. Two samples (S912; S1036) were positive for both *Salmonella* sp. and *V. fluvialis*. All of the 47-fish examined from six separate bags were clean with no visible mud, food items or vegetation (Table 1).

#### 3.2.1. *Vibrio fluvialis*

*Vibrio fluvialis* grew on SAL from 10/47 swabs cultured and a mean of 13.9 cfu/g (Table 2). All samples were identified using RNF Plus. Samples S878 and S879 had a small percentage overlap and S916 one contraindication all for *Plesiomonas shigelloides* in the RNF Plus system. DNAse activity and negative Lysine decarboxylase (PC: CM0308S) confirmed samples as *V. fluvialis* (Table 3).

#### 3.2.2. *Salmonella* sp.

*Salmonella* sp. was cultured on SAL from 6/47 swabs and had a mean of 41.6 cfu/g (Table 2) and was confirmed as *Salmonella* sp. using the *Salmonella* Latex Test in 100% of the samples tested. *Salmonella* sp. from Species B showed variable antimicrobial patterns of resistance (Table 3).

#### 3.2.3. *Micrococcus* sp. and *S. aureus*

*Micrococcus* sp. was cultured from 3/47 swabs with a mean of 4.6 cfu/g and *S. aureus* was cultured from one swab and 1.0 cfu/g only was observed (Table 2). *Micrococcus* sp. and *S. aureus* identification followed steps described in Table 3. The detailed antimicrobial patterns of resistance are included in Table 4. It is noted that *Staphylococcus aureus* was resistant to all antimicrobials tested. 

### 3.3. Associations between Identified Biosecurity Breeches (Mud, Food Items or Vegetation) and Bacteria

Insufficient data were available for several predictor variables to confidently report associations with presence or absence of bacteria species, due to zero counts for at least one of the predictor levels. For example, *Staphylococcus aureus*, *V. fluvialis* and *Salmonella* species were only present in specimens where mud was absent. Similarly, these bacteria species were only present at low levels in specimens where components of a natural diet or vegetation were absent. *Rhizobium* (syn. *Agrobacterium*) *radiobacter* and *Pseudomonas* spp. were observed in Sp. A, but not in Sp. B. In contrast, *V. fluvialis* was reported in Sp. B but not Sp. A (Table 5).

Where adequate data were available, univariable logistic regression identified significant associations between predictor variables and bacteria occurrence (Table 6). The identification of mud in specimens was significantly associated with the presence or absence of *Pseudomonas* spp. Where mud was recovered from specimens, the recovery of *Pseudomonas* spp. was 7.58 (CI 2.95, 20.72) more likely (*p* < 0.001). The presence of environmental contaminants was also associated with an increased likelihood of *Pseudomonas* spp. contamination. Both the presence of natural diet components (OR 8.46; CI 3.37, 22.93), and the presence of vegetation (OR 13.10; CI 3.10, 90.26) were significantly associated with recovery of *Pseudomonas* spp. (*p* < 0.001).

Fish species was significantly associated with some bacteria species. *Pseudomonas* spp., *R. radiobacter* and *C. testosteroni* were only identified in Sp. A and *V. fluvialis* in Sp. B. One sample of *S. aureus* was identified in each of the Sp. A and Sp. B samples. The occurrence of *Micrococcus* sp. was more likely (OR 11.84; CI 3.75, 52.80; *p* < 0.05) in Sp. A and *Salmonella* sp. less likely (OR, 0.09; CI 0.005, 0.61; *p* < 0.001) in Sp. A than in Sp. B.

## 4. Discussion

The present study aimed to identify if selected bacteria are present on the skin or outer surface of two edible fish products from Countries 20 and 22, and to explore associations between bacteria isolated and a set of explanatory variables which were based on the supply chain breaches identified in Williams, et al. [81]. Zoonotic bacteria, *S. aureus* and *Salmonella* sp., *C. testosteroni* with zoonotic potential and *R. radiobacter*, *Micrococcus* sp., *P. mendocina* and *P. pseudoalcaligenes* implicated in opportunistic cases of human infection were cultured from the skin/outer surface and the ‘bag water’ of edible imported Sp. A (Country 22). Zoonotic species *Vibrio fluvialis*, *Salmonella* sp. and *S. aureus* and the opportunistic *Micrococcus* sp. were identified from Sp. B fish (Country 20). Univariable logistic regression analysis demonstrated that *Pseudomonas* spp. were significantly associated with the presence of mud, natural diet components and presence of vegetation. This infers that these types of breaches to importation regulations have potential to introduce zoonotic/opportunistic bacteria into the Australian food chain. Species A (Country 22) had the greatest diversity of bacteria. Antimicrobial sensitivity testing demonstrated that for some samples of *Salmonella* sp. and *S. aureus*, resistance was demonstrated to all antimicrobials tested. This study showed that the outermost surface of imported frozen edible freshwater fish has potential to contaminate hands, work surfaces, kitchen sponges and other kitchen utensils with bacteria which are not only a human health concern but are capable of forming a persistent biofilm. All species or genera of bacteria identified from Sp. A and Sp. B have been implicated in cases of community and/or nosocomial infection, including a greater vulnerability to infection demonstrated in the immune compromised. In this study only a small area of fish skin was swabbed and therefore it is considered that there is significant contamination of the skin of Sp. A with *Pseudomonas* species (*P. mendocina* and *P. pseudoalcaligenes*), *C. testosteroni* and *Micrococcus* sp. and with *V. fluvialis* and *Salmonella* in Sp. B. 

According to Australian importation policy, for Sp. A and Sp. B, all consignments must be clean and be free of ‘live insects, seeds, soil, mud, clay, animal material (such as faeces), plant material (such as straw, twigs, leaves, roots, bark) and other debris prior to arrival into Australian territory’ [112]. There was a significant association between mud, natural diet and vegetation and the presence of *Pseudomonas* spp. in specimens of Sp. A. As an environmental bacterium, the presence of *Pseudomonas* spp. in mud seems likely, however the ability to form a biofilm on environmental debris [113,114] may explain the association with natural diet, vegetation and other debris observed in bags of contaminated fish. Whilst this association was identified for *Pseudomonas* spp., other environmental bacterium which are associated with mud and can adopt a biofilm existence, such as *C. testosteroni* and *R. radiobacter*, as identified in this study are also a contamination concern on the aforementioned debris.

The natural diet identified in Sp. A in the present study consisted of Lymnaeid and Planorbid snails both of which prefer lentic environments with stagnant water [115]. Species A were imported into Australia from a country (Country 22) which, according to Wendling, et al. [83], has a country score of 0/100 for ‘Wastewater Treatment’ and less than ‘30/100′ for ‘Water and Sanitation’ (lower scores representing least favourable). *Pseudomonas* spp. are associated with polluted and eutrophic freshwater [116,117]. It seems mud and this species of bacteria are synonymous in fish from countries with poor environmental controls in FF and the *Pseudomonas*: mud relationship closely associated with fish habitat and feeding presences. Consequently, failure to comply with Australian importation guidelines has potential to introduce *Pseudomonas* spp. into the Australian food chain.

As Sp. A and Sp. B are not ‘risk’ foods they are subject to a reduced rate of inspection on entry into Australia. Five percent of samples may be randomly selected for label/visual inspection or sent for analysis to determine if the product is compliant with Australian food safety standards. However, if the exporting country has a compliance agreement with Australia, Sp. A and Sp. B may not be inspected on entry providing the product is accompanied by a declaration of safety compliance ‘*under the supervision of the competent authority and/or systems approved by the competent authority*’ [76,77,112,118,119]. Whilst microbiological limits are set for some ‘risk’ seafood (shellfish, prawns, crabs) imported into Australia in the FSANZ [120], other than *L. monocytogenes* [78], the zoonotic bacteria identified in the present study may not be isolated from Sp. A or Sp. B if tested on entry into Australia. According to the international health standards established by Codex Alimentarius [121] the ‘Code of Practice for Fish and Fishery Products’ (CXC 52-2003) states ‘*to remove foreign debris and reduce bacterial load prior to gutting*’ (whole fish) and glazing dips should be replaced periodically ‘*to minimize the bacterial load*.’ In section 22.1.2 of the ‘code of practice’, ‘Reception of frozen products at retail’ ‘*evidence of filth or contamination*’ at the retail level is an ‘*unlikely*’ hazard [122], although high levels of filth and contamination were identified in Sp. A in the current study. 

Of the zoonotic bacteria identified in the present study, *Salmonella* species and *S. aureus* are the only bacteria included as a potential hazard for the fish examined in CXC 52-2003 [123]. *Vibrio* spp. is mentioned, which ‘*can be controlled by thorough cooking and preventing cross-contamination of cooked products*’ and ‘*indigenous pathogenic bacteria, when present on fresh fish, are usually found in fairly low numbers, and food safety hazards are insignificant where products are adequately cooked prior to consumption*’. The Codex code of practice for fishery products fails to recognise pathogenic bacteria on the surface of uncooked thawed imported fish as a potential human health hazard. Contamination of hands, cutting surfaces and kitchen utensils prior to cooking is a significant threat and one which has been described in published literature [124,125,126,127,128]. Far from indigenous bacteria occurring in low numbers in the present study, bacteria *C. testosteroni*, *Pseudomonas* spp., *V. fluvialis* and *Micrococcus* sp. had a high RR. *Salmonella* sp. had a medium RR which may indicate that water used for glazing fish fillets during processing is not compliant with CXC 52-2003 standards [123].

Species of fish was also significantly associated with certain species of bacteria. Whilst Sp. A had many breaches according to Australian edible fish importation requirements, Sp. B appeared clean and well glazed, indicating separate bacterial hazards exist for both groups of fish from different regions and some of these hazards are not observable. Therefore, targeting high risk fish for bacterial contamination on entry into Australia seems achievable. In addition, in compliance with ‘*The General Agreement on Sanitary and Phytosanitary Measures*’ [121], developed nations are encouraged to help developing nations to reach food safety compliance. Inexperienced or untrained labour may be responsible for the variable processing standard of Sp. A, and this may be an area where Australia could offer assistance.

*Micrococcus* sp. was isolated from both Sp. A and Sp. B. *Micrococcus* spp. are common across aquatic and terrestrial ecosystems, the skin of humans, other endothermic animals [129] and frequently contaminate food. *Micrococcus* spp. have been identified at farm level in cultured Nile tilapia (*Oreochromis niloticus*) [130,131], striped catfish (*Pangasius hypophthalmus*) [56], isolated during filleting and trimming of cultured striped catfish [56] and cassava fish (*Pseudotolithus* sp.) [132]. *Micrococcus* spp. contamination has been described in preserved [133] and fermented fish at the retail level in developing countries [134,135]. Pathogenicity of *Micrococcus* spp. has been multiply described in cases of opportunistically acquired community human infection [69,136,137,138,139,140,141,142], demonstrating the potential for contaminated imported FP to be a health risk to consumers.

In the present study, *Micrococcus* spp. was resistant to Cefovecin and Vancomycin. Traditionally, *Micrococcus* spp. are sensitive to a broad range of antimicrobials [138,143,144]; however, as in this study, resistance to Vancomycin has been described in some clinical cases of human infection [140,145]. The occurrence of *Micrococcus* in edible fish imported to Australia deserves greater attention. 

*Comamonas testosteroni* was identified in Sp. A only. This bacterium is common in the environment [146,147] but has a demonstrated prevalence in anthropogenically impacted environments [148], including polluted lakes/regions, agricultural land, urban sewerage [149] and farm polluted aquifers [116]. *Comamonas testosteroni* has been identified in cultured common carp (*Cyprinus carpio* & *Carassius auratus*) [150], Nile tilapia and African sharp-tooth catfish (*Clarias gariepinus*) [151]. Community transmission has occurred via minor abrasions from fish and contaminated food [147,152,153]. Smith and Gradon [152] reported a case of human infection from *C. testosteroni* speculated to have been transferred from diseased tropical fish and the zoonotic potential of this species requires clarification. Due to an ability to utilise nitrate as a substrate, this aerobic species can exist in anaerobic conditions and adopt a biofilm existence [154]. This species has been multiply reported in opportunistic community acquired invasive infections [68,147,153] and imported FP contaminated with this bacterium may be a consumer risk.

In the present study *C. testosteroni* was sensitive to all antimicrobials, which is supported by sensitivity patterns in clinical cases of human infection [155,156,157]. However, resistance to some commonly used antimicrobials has recently been observed [146,147,156,158] in human clinical cases and the high RR of contamination on the outer surface of imported edible Sp. A may require further attention.

Both *P. mendocina* and *P. pseudoalcaligenes* were identified from Sp. A. *Pseudomonas* spp. are environmental bacteria which demonstrate an extremely high degree of genetic and physiological adaptability [159]. Member species increase yearly [160]. *Pseudomonas* sp. have been isolated from cultured tilapia (*Oreonchromis* spp.) and pond water [161] and *P. pseudoalcaligenes* from water at a freshwater prawn hatchery [162]. *Pseudomonas mendocina* is able to form a biofilm on environmental microplastics [113,114], is associated with polluted/eutrophic freshwater [116,117], and demonstrates increased antimicrobial resistance in manure treated soils [163]. Cultured freshwater fish are frequently described contaminated with *Pseudomonas* spp. at the farm level [130,131,161,164,165], during processing [56] and in retail fresh/frozen cultured freshwater fish [55,166,167]. *Pseudomonas mendocina* has been isolated from processed fish products [168,169]. *Pseudomonas pseudoalcaligenes* has been isolated from cultured common carp for retail sale [170] and was the causative agent of mass death in cultured Nile tilapia [171].

No published literature has identified *Pseudomonas* genus as zoonotic. However, the many cases of community acquired opportunistic infection of humans following contact with environmental sources [70,172,173,174] demonstrates the potential of contaminated imported FP to be a health risk to consumers. The case of human infection from a pet bird in which *P. mendocina* was cultured from the bird’s throat and drinking water also supports the further investigation of this species as a zoonosis [175].

In the present study *Pseudomonas* sp. demonstrated an expected pattern of antimicrobial sensitivity according to the CLSI [105]. Resistance to a broad range of antimicrobials has also been reported [176] in hospital and other settings [177]. *Pseudomonas* spp. are able to exchange genes (conjugation/horizontal gene transfer) which may favour increased virulence and spread between species [178].

*Rhizobium radiobacter*, identified in three samples of Sp. A, is a pathogen that can cause disease in both humans and plants. This species is widely distributed in soil as well as hospital environments [111], although reports of *R. radiobacter* contaminating fish are absent from literature. In the present study one sample came from a bag where vegetation, mud and natural diet were observed and another where mud and natural diet were both present. The preferred habitat of Sp. A is stagnant waters in muddy streams [179] which may explain the presence of this bacterium. *Rhizobium radiobacter* in the present study grew during incubation and produced small, bright orange, non-haemolytic colonies on MacConkey agar and it is possible this may have been the clone described in cases of human infection. Chanza, et al. [111], in a report of a pulmonary abscess in a Spanish 64-year-old male, notes growth on MacConkey agar as ‘bright orange, non-haemolytic colonies.’ The ability to grow at high temperature is an emergent trait in this bacterium. According to Hélène, et al. [180], a primary condition for colonization and pathogenicity is living in close proximity to the bacteria. It was suggested by Coenye, et al. [181] that *R. radiobacter* has potential for person to person transmission, therefore investigation into the spread between consumers via contaminated fish seems warranted. As a biofilm inhabitant, this bacterium contaminating FP may lead to colonisation of kitchen items used during preparation/clean up.

Human infection with *R. radiobacter* is described as rare in published literature, however Table 7 shows 23 cases. In many cases of human infection, contact with vegetation [72,182] and soil [72,73,183] is described. *Rhizobium radiobacter* is associated with invasive infections, particularly in the immunodeficient and children [111,184,185,186,187,188] and this bacterium present on FP may be an unacceptable risk.

In the present study *R. radiobacter* was sensitive only to Ciprofloxacin. This bacterium has been identified as sensitive to Ciprofloxacin in a number of studies [184,239,240], as expected, resistant to Vancomycin, and a large range of other antimicrobials [241,242]. According to Edmond, et al. [241], many bacterial species found in soil are innately resistant to antimicrobials and *R. radiobacter* has been demonstrated to ‘produce multiple antibiotic-inactivating enzymes’.

*Vibrio fluvialis* was only isolated from imported Sp. B fillets in the present study. The halophilic *Vibrio fluvialis* has traditionally been recovered from marine/estuarine environments, however *V. fluvialis* is now reported from cultured freshwater fish [61,62,243,244] and is considered an emerging foodborne pathogen [245]. *Vibrio fluvialis* has been isolated from river mud samples in Bangladesh [246], human stools, and faecally contaminated [247] as well as treated effluents systems [248] and is able to form a biofilm on environmental microplastics [249]. *Vibrio fluvialis* has been isolated at the farm level from cultured Nile tilapia and North African catfish [243,250,251], cultured African snakehead (*Parachanna africana*) and bagrid catfish (*Chrysichthys nigrodigitatus*) [244], common carp [252], rohu (*Labeo rohita*) [253,254], fish farm water [61] and in retail frozen/fresh fish [58,255].

Large outbreaks of cholera-like diarrhoea from *V. fluvialis* have been reported after consumption of contaminated fish/freshwater [225,227,230], although many outbreaks have an unknown origin [233]. *Vibrio fluvialis* was the causative agent for haemorrhagic cellulitis leading to amputation [231] and a urinary tract infection [238] from heavily contaminated water.

In the present study, all samples were resistant to Colistin, Cephalothin and Vancomycin, which has been previously described in published literature [103,232,256,257] (ref. [257] is a preprint). *Vibrio fluvialis* has been demonstrated resistant to a wide variety of common antimicrobials and it is considered that a rapid expansion in resistant strains [258] has occurred. 

Two samples, one each from Sp. A and Sp. B, were identified as *S. aureus*, with one sample being resistant to all antimicrobials tested and considered to be MRSA. *Staphylococcus aureus* is found in the environment and the nose and skin of healthy humans [259]. This bacterium can be spread by direct contact between people and/or contact with contaminated objects [260], and *S. aureus* on the outer surface of edible fish has potential to contaminate hands/work surfaces or be transmitted between family members/pets. *Staphylococcus* spp. and *S. aureus* have been isolated at the farm level from cultured tilapia [130,161,261], striped catfish [56,166], during filleting/trimming of cultured striped catfish [56], and in retail frozen/fresh/fermented fish [134,166,167,262].

Coagulase positive staphylococcal food poisoning is a dominant cause of foodborne illness globally [263] and one of the most common bacterial infections described in other clinical conditions [264,265,266,267,268,269,270,271,272,273,274,275,276]. 

In the present study *Salmonella* sp. was identified from Sp. A and Sp. B, although not all species belonging to this genus are pathogenic to humans [277]; the testing of edible food, including fish, which may be carried out on entry to Australia only lists the genus *Salmonella*, as does Codex ‘Code of Practice for Fish and Fishery Products’ (CXC 52-2003) [123]. *Salmonella* spp. are found in the digestive tract of a broad range of animals. Humans, cows, pigs, and chickens are recognised carriers [278] and may contaminate food and water sources through the excretion of this bacterium in faeces [279]. *Salmonella* is spread from consumption of contaminated food, but may also be passed from person-to-person [280,281] and between animals and people [282]. In regions where the intensive freshwater FF system is used particularly in conjunction with poor sanitation standards, *Salmonella* will be prevalent in both the water and the fish raised in these facilities. 

*Salmonella* sp. has been isolated from the waters where cultured catfish [283], Mozambique tilapia (*Oreochromis mossambicus*) and catfish [63] are raised. In Vietnam, it is considered that the important contamination sources in FF are ‘*livestock and non-livestock-reservoirs, such as faecal-contaminated pond water, fish feed, birds, amphibians and reptiles*’ [284]. *Salmonella* sp. has been isolated at the farm level from tilapias [130,261,285] and African sharp-tooth catfish [63], in retail catfish/other cultured freshwater fish [59,262,286] and in an array of imported seafood [60,287,288,289,290,291]. In the USA, the incidence of *Salmonella* contamination in imported raw fish was 12.2%, with Vietnam having the highest by country basis [60].

Salmonellosis, which has increased markedly in the last 50 years, is associated with gastroenteritis and many other types of systemic disease [60,292]. Seafood borne salmonellosis outbreaks have occurred in the United Kingdom [223,293], Germany [224], the European Union and the USA, although according to Heinitz, et al. [60] salmonellosis is significantly under reported.

In the present study, *Salmonella* sp. sample S469 showed a total pattern of resistance for all antimicrobials tested which is supported in recent reports of multi antimicrobial resistance in *Salmonella* sp. samples from edible fish and other food products [294,295,296]. 

Seafood related salmonellosis and staphylococcal infection has been recognised globally for some time; however, the emergence of new human pathogens from environmental bacteria, previously not considered a human health threat, is of concern. The international trade in cultured edible fish cannot be disregarded as a significant circulator and propagator of these environmental zoonotic/non-zoonotic bacteria and antimicrobial resistant strains into regions where they have not previously been recognised.

A limitation of the present study was the small sample size and lack of more sophisticated methods of identification for bacteria, due to limitations in time and resources available. It is considered that future studies could use MALDI-TOF^®^, mass spectrometry for identification of bacteria, and molecular and phenotypic methodology to characterize antimicrobial patterns of resistance.

## 5. Conclusions

In conclusion, zoonotic bacteria, many with a high RR, were identified contaminating frozen edible Sp. A and Sp. B imported into Australia. It is suggested that further study, using a greater sample size, may support the development of new food safety guidelines for Australian consumers for certain types of imported freshwater fish. It is also suggested that Australia could provide some countries, which produce farmed edible fish, with support to reach greater food safety compliance.

## Figures and Tables

**Figure 1 foods-12-01288-f001:**
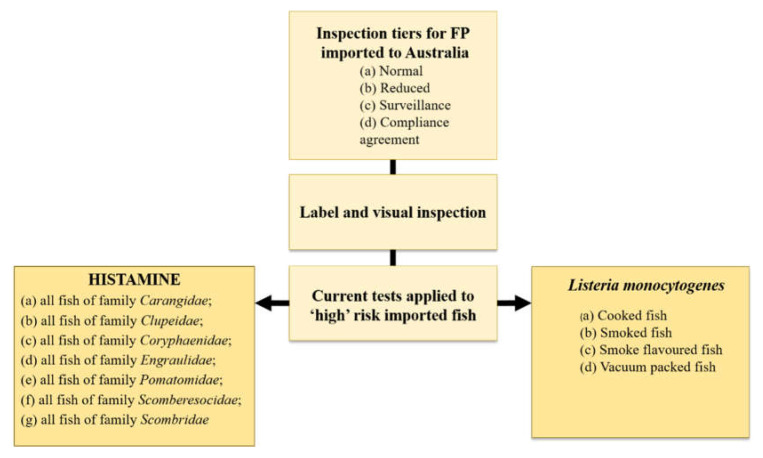
**The current tests applied to high risk imported fish on entry to Australia.** *Listeria monocytogenes* at present is the only additional test for bacteria applied to imported fish on entry to Australia. Original figure based on information in the Imported Food Control Order, 2020.

**Table 1 foods-12-01288-t001:** Processing observations of fish and packaging in this study and corresponding sample numbers.

Species A (Country 22) Packaging Observations	Sample ID	Species B (Country 20) Packaging Observations	Sample ID
**BAG 1** Gills and lower jaw removed in all fish. Intestine still in most. Natural diet with cricket legs. Mud in intestine/gills. Total of 500 g	S334-Bag water S335-S345	BAG 1 Clean and individual fillets well glazed. Total of 1 kg	S878-Bag water S879-S885
**BAG 2** Natural diet in all. Intestine in most. No brains in any. Bag water clear. Quite well cleaned. Total of 500 g	S431-S441 S446-Bag water	BAG 2 Clean and individual fillets well glazed. Total of 1 kg	S894-Bag water S895-S900
**BAG 3** Moderately clean. No brains in any. Total of 500 g	S464-S474 S476- Bag water	BAG 3 Clean and individual fillets well glazed. Total of 1 kg	S908-Bag water S909-S915
**BAG 4** Fish all very dirty. Gills full of mud and debris. Intestine/gills all in. Remnants of brains. Debris in bag water, bits of crustacean shells and vegetation. Natural diet. Total of 500 g	S649-Bag water S650-S660	BAG 4 Clean and individual fillets well glazed. Total of 1 kg	S1007-Bag water S1008-S1015
**BAG 5** Natural diet. Vegetation. Filthy intestine and gills in all. Mud in gills and full of debris. Bits of brain left in. Total of 500 g	S681-Bag water S686-S691	BAG 5 Clean and individual fillets well glazed. Total of 1 kg	S1023-Bag-water S1046-S1052
**BAG 6** Very well cleaned and gutted. Most of brains gone. No natural diet. Total of 500 g	S699-Bag water S700-S709	BAG 6 Clean and individual fillets well glazed. Total of 1 kg	S1031-Bag water S1032-S1038

**Table 2 foods-12-01288-t002:** **List and number of bacteria found between the two species of imported fish colonies per gram (cfu/g).** Bag water from Sp. A (n = 6) and Sp. B. (n = 6) has been included in the fish overall number. Mean not calculated when only one fish contaminated *.

Fish Number (n =)	Bacterial Genera or Species	Number of Contaminated Fish	CFU Mean Contaminated Fish cfu/g	Recovery Rate (%)
**Sp. A from Country 22 (n = 66) from six separate bags**	*Micrococcus* sp.	32	50.0	48.5
	*Staphylococcus aureus*	1	27.0 *	1.5
	*Comamonas testosteroni*	27	61.7	40.9
	*Pseudomonas* sp. (including *P. mendocina* and *P. pseudoalcaligenes*)	34	104.0	51.5
	*Salmonella* sp.	1	32.0 *	1.5
	*Rhizobium radiobacter*	3	7.6	4.5
**Sp. B from Country 20 (n = 47) from six separate bags**	*Salmonella* sp.	6	41.6	12.7
	*Micrococcus* sp.	3	4.6	6.4
	*Staphylococcus aureus*	1	1.0 *	2.1
	*Vibrio fluvialis*	10	13.9	21.3

**Table 3 foods-12-01288-t003:** Growth media and tests for identification of bacteria.

Growth Media	Bacterium	Colony Growth on Media	Gram Stain	Biochemical Tests	Other Confirmatory Tests	Diagnostic Anti-Microbial Resistance
**Species A**						
**CFC**	*Pseudomonas* sp.	Rapid growing mucoid, slightly pigmented, pale metallic green, round, convex	Gram negative rods with flagellar		Catalase + Oxidase + Coagulase −	Cefovecin Cefoxitin Vancomycin
	*Pseudomonas mendocina*	Rapid growing mucoid, slightly pigmented, pale metallic green, round, convex	Gram negative rods with flagellar	Confirmed RNF Plus n= 6/15	Glucose oxidation	
	*Pseudomonas pseudoalcaligenes*	Rapid growing mucoid, slightly pigmented, pale metallic green, round, convex	Gram negative rods with flagellar	Confirmed RNF Plus n= 9/15	Glucose alkalisation	
**BP**	*Micrococcus* sp.	Medium sized, shiny black, convex	Gram-positive cocci, paired/and or groups of four		Coagulase slide/tube – Catalase + Oxidase +	
	*Staphylococcus aureus*	Small, black, dry to mucoid, slightly raised but flat, distinct yellow halo surrounding colonies	Gram-positive cocci, clustered/bunched		Coagulase slide/tube + Catalase + Oxidase −	
**TSC**	*Comamonas testosteroni*	Medium sized, mucoid, convex, dark cream to light brown in the centre of colony	Gram negative, small rod shaped	Confirmed RNF Plus n = 12/12	Glucose alkalisation Lactose-fermentation − Catalase + Oxidase +	
**MH**	*Rhizobium radiobacter*	Pale yellowish, slightly dry, round to slightly irregular colonies	Gram negative, rod shaped	Confirmed RNF Plus n = 3/3	Lactose-fermentation + Catalase + Oxidase +	Vancomycin
**SAL**	*Salmonella* sp.	Very small, convex, slightly mucoid colonies	Gram stain negative, rod shaped	Confirmed *Salmonella* Latex Test n = 1/1	Catalase + Oxidase +	
**Species B**						
**SAL**	*Vibrio fluvialis*	Large purple flat dense mucoid on SAL. MH characteristic growth	Gram stain negative, small rod-shaped some curved	Confirmed RNF Plus n = 10/10	Catalase + Oxidase + DNAse activity + Lysine decarboxylase −	Colistin Cephalothin
**SAL**	*Salmonella* sp.	Very small, convex, slightly mucoid colonies	Gram stain negative, rod shaped	Confirmed *Salmonella* Latex Test n = 6/6	Catalase + Oxidase +	
**BP**	*Micrococcus* sp.	Medium sized, shiny black, convex	Gram-positive cocci, paired/and or groups of four		Coagulase slide/tube − Catalase + Oxidase +	
	*Staphylococcus aureus*	Small, black, dry to mucoid, slightly raised but flat, distinct yellow halo surrounding colonies	Gram-positive cocci, clustered/bunched		Coagulase slide/tube + Catalase + Oxidase −	

**Table 4 foods-12-01288-t004:** **Percentage of antimicrobials susceptible (S), intermediate (I) or resistant (R).** From *Sp. A and from #Sp. B. CVN (Cefovecin), FOX (Cefoxitin), CIP (Ciprofloxacin), FOS (Fosfomycin), VAN (Vancomycin), Colistin (CT) and Cephalothin (KF).

Bacteria	CVN			FOX			CIP			FOS			VAN			CT/KF		
%	S	I	R	S	I	R	S	I	R	S	I	R	S	I	R	S	I	R
*C. testosteroni*	100	-	-	100	-	-	100	-	-	100	-	-	100	-	-	-	-	-
*Micrococcus* sp. *	-	-	100	100	-	-	100	-	-	100	-	-	-	2.8	97.2	-	-	-
*Micrococcus* sp. #	-	-	100	100	-	-	100	-	-	100	-	-	-	-	100	-	-	-
*Pseudomonas* sp.	-	-	100	-	-	100	94.2	5.8	-	100	-	-	-	-	100	-	-	-
*P. mendocina*	-	-	100	-	-	100	100	-	-	100	-	-	-	-	100	-	-	-
*P. pseudoalcaligenes*	-	-	100	-	-	100	88.9	11.1	-	100	-	-	-	-	100	-	-	-
*R. radiobacter*	-	-	100	-	-	100	100	-	-	-	-	100	-	-	100	-	-	-
*Salmonella* sp. *	-	-	100	-	-	100	-	-	100	-	-	100	-	-	100	-	-	-
*Salmonella* sp. #	-	16.6	83.3	-	-	100	50	50	-	-	-	100	-	-	100	-	-	-
*S. aureus* *	-	-	100	100	-	-	100	-	-	100	-	-	-	-	100	-	-	-
*S. aureus* #	-	-	100	-	-	100	-	-	100	-	-	100	-	-	100	-	-	-
*Vibrio fluvialis*	70	20	10	-	-	100	90	10	-	80	10	10	-	-	100	-	-	100

**Table 5 foods-12-01288-t005:** **Occurrence of bacteria species by predictor variables.** Bracketed numbers in ‘Occurrence’ column show the total count of fish from Sp. A (n = 60) and Sp. B. (n = 41). Bag water not included in statistical evaluation. Predictor variables based on supply chain breaches identified in Williams et al. (2021).

Species	Predictor Variable	Term	Occurrence
*S. aureus*	Mud	Mud absent	2.74% (73)
		Mud present	0% (28)
	Nat Diet	Components of natural diet present	3.23% (62)
		Components of natural diet absent	0% (39)
	Veg	Vegetation recovered	2.22% (90)
		Vegetation absent	0% (11)
*Salmonella* sp.	Mud	Mud absent	9.59% (73)
		Mud present	0% (28)
	Nat Diet	Components of natural diet present	11.29% (62)
		Components of natural diet absent	0% (39)
	Veg	Vegetation recovered	7.78% (90)
		Vegetation absent	0% (11)
*V. fluvialis*	Mud	Mud absent	9.59% (73)
		Mud present	0% (28)
	Nat Diet	Components of natural diet present	9.59% (73)
		Components of natural diet absent	0% (28)
	Veg	Vegetation recovered	11.29% (62)
		Vegetation absent	0% (39)
	Fish	Pangasiidae	7.78% (90)
		Channidae	0% (11)
*R. radiobacter*	Fish	Pangasiidae	0% (41)
		Channidae	5.08% (59)
*Pseudomonas* spp.	Fish	Pangasiidae	0% (41)
		Channidae	53.33% (60)

**Table 6 foods-12-01288-t006:** **Associations between predictor variables and bacteria. Odds ratios calculated using the first level as a reference.** Bracketed numbers in ‘Occurrence’ column show the total count. Sp. A (n = 60) and Sp. B. (n = 41). Bag water not included in statistical evaluation. Predictor variables based on supply chain breaches identified in Williams et al. (2021).

Bacteria Species	Predictor	Levels	Occurrence (N)	Odds Ratio (95% CI)	*p* Value
*Pseudomonas* spp.	Mud	Mud absent	19.18% (73)		<0.001
		Mud present	64.29% (28)	7.586 (2.954, 20.726)	
	NatDiet	Nat diet (0)	14.52% (62)		<0.001
		Nat diet (1)	58.97% (39)	8.465 (3.374, 22.939)	
	Veg	Veg (0)	25.56% (90)		<0.001
		Veg (1)	81.82% (11)	13.109 (3.104, 90.265)	
*Salmonella* sp.	Fish	Sp. B	14.63% (41)		0.010
		Sp. A	1.67% (60)	0.099 (0.005, 0.611)	
*Micrococcus* sp.	Fish	Sp. B	7.32% (41)		<0.001
		Sp. A	48.33% (60)	11.849 (3.758, 52.803)	

**Table 7 foods-12-01288-t007:** **Cases of human infection associated with zoonotic bacteria identified in this manuscript**. N/S indicates not stated in cited publication.

Bacteria Identified	Source of Infection	Patient Details	Clinical Notes	Region Infection Was Identified
** *Comamonas testosteroni* **	Central venous catheter	54-year-old female	Bacteraemia	Washington, USA [155]
	Intrauterine device	29-year-old female	Peritonitis	Turkey [189]
	Unknown	50-year-old male	Purulent meningitis	Turkey [190]
	Contamination of the distilled water used to decontaminate aspiration catheter possibility	10-year-old male	Pneumonia leading to death	Turkey [158]
	Community acquired possibly from small skin wounds	49-year-old male	Endocarditis	USA [68]
	Unknown but possible hospital acquired	19-year-old female and 10-year-old male	Bacteraemia	Iran [156]
	Community acquired in situ colostomy	65-year-old female	Gastroenteritis	India [146]
	In dwelling central venous catheter	75-year-old female	Bacteraemia	France [191]
	Indwelling catheter	64-year-old female	Bacteraemia leading to death	Israel [192]
	Community acquired possibly right shoulder rotator cuff tendinitis.	80-year-old female	Bacteraemia	USA [193]
	Incidental dislocation of an intrauterine device	4-year-old female	Peritonitis	Italy [194]
	Community acquired	1-year old female	Sepsis following gastroenteritis	Oman [195]
	Community acquired from handling diseased tropical fish	89-year-old male	Bacteraemia	Unknown [152]
	Community acquired	14-year-old male and one child	Appendicitis	Turkey [196]
	Contaminated food and water speculated	46-year-old female	Bacteraemia	India [147]
	Community acquired from leg injury from a fish fin	54-year-old male	Leg cellulitis and bacteraemia	Taiwan [153]
	Community acquired	73-year-old male	Intra-abdominal infection	Taiwan [153]
	Hospital acquired	68-year-old male	Bacteraemia leading to death	Turkey [197]
** *Micrococcus* ** **species**	Community transmission	26-year-old female	Micrococcal pneumonia leading to death	Netherlands [136]
	Community transmission via Broviac catheter	13-year-old female	Death	N/S [137]
	Community acquired through eye injury	30-year-old	*Micrococcus* endophthalmitis	USA [69]
	Community transmission	1-year old male	Micrococcal meningitis with neurological sequalae	India [138]
	Community acquired from peritoneal dialysis catheters	6 patients	N/S	United Kingdom [139]
	Unknown but possible introduced during previous surgery	74-year-old female	Aortic valve endocarditis, recurring septicaemia	Greece [140]
	Unknown	2 patients	Septic arthritis	United Kingdom [198]
	Community acquired	8 patients	Pitted keratolysis of the foot	United Kingdom [199]
	Community transmission	22-year-old male	Micrococcal pneumonia leading to death	Spain [141]
	Community transmission via tooth cavity	37-year-old female	Intracranial suppuration	Malaysia [142]
	Community transmission	69-year-old male	Micrococcal pneumonia	Canada [200]
	Via catheter a possibility	16-year-old female	Bacteraemia	Germany [201]
** *Pseudomonas mendocina* **	Community acquired via thorn pricks of hands	63-year-old male	Infective endocarditis	Argentina [70]
	Community acquired suspected via trivial wound	65-year-old male	Lumbar spondylodiscitis	Taiwan [172]
	Community acquired via foot wound	34-year-old male	Septic arthritis	Singapore [173]
	Suspected related to the open wound	64-year-old male	Bacteraemia	USA [174]
	Likely community acquired via a central venous catheter	72-year-old male	Bacteraemia	New Orleans USA [202]
	Unknown	64-year-old male	Bacteraemia	United Kingdom [203]
	Community acquired	86-year-old female	Atypical fracture of the femur.	Unknown [204]
	Community acquired	55-year-old male	Meningitis	Taiwan [205]
	Unknown	66-year-old female	Meningitis	Taiwan [205]
	Hospital acquired	79-year-old male	Meningitis	Taiwan [205]
	Community acquired	79-year-old female	Meningitis	Taiwan [205]
	N/S	28-year-old woman	Infective endocarditis	Denmark [206]
	Unknown	63-year-old man	Peritonitis	Portugal [207]
	Community acquired	36-year-old male	Native mitral valve endocarditis	Turkey [208]
	Community acquired from drinking water of pet bird	31-year-old male	Sepsis	Israel [175]
	Leg wounds likely source	57-year-old male	Native mitral valve endocarditis	USA [71]
	Unknown	79-year-old female	Endocarditis	France [209]
** *Pseudomonas pseudoalcaligenes* **	Dialysis portal	29-year-old female	Peritonitis	New York, USA [210]
	Unknown	17-year-old pregnant female	Stillborn foetus	USA [211]
** *Rhizobium radiobacter* **	Community transmission via central venous catheter	6-year-old female and 10-year-old male	Bacteraemia	Texas, USA [212]
	Community acquired after leaf contacted eye	42-year-old female	Corneal abscess	Romania [182]
	Hospital acquired via central venous catheter	34 weeks premature baby	Bacteraemia	India [213]
	Hospital acquired likely via implicated central venous catheter	2-year-old male	Bacteraemia	Macedonia [214]
	Normal environment	64-year-old male	Pulmonary abscess	Valencia, Spain [111]
	Hospital acquired from unknown source	19-year-old male	Bacteraemia	Greece [184]
	Via central venous catheter	42-year-old female	Bacteraemia	Belgium [215]
	Community acquired	80-year-old male	Bacteraemia	Belgium [215]
	Community acquired via transcutaneous catheter	3.5-year-old male	Bacteraemia	Texas, USA [185]
	Community acquired via contact lens whilst bathing and swimming	26-year-old female	Keratitis	China [216]
	Hospital acquired	2-year-old, 37-year-old females and 21, 46, 53, 59, 79-year-old males	Catheter-related bacteraemia	Taiwan [186]
	Community acquired	61-year-old female	Pneumonia	Taiwan [186]
	Central venous catheter community and hospital acquired	73-year-old female and 53-year-old male	Bacteraemia	France [217]
	Community acquired plant and soil via peritoneal dialysis	41-year-old male	Peritonitis x 2 leading to death	USA [72]
	Via peritoneal dialysis catheter community acquired	43-year-old male	Peritonitis	China [218]
	Community acquired via a peritoneal dialysis catheter	63-year-old male	Relapsing peritonitis	Spain [187]
	Community acquired	54-year-old male	Peritonitis	India [219]
	Hospital acquired via ventilator humidifier water and water-chamber of ventilator	6 hospitalised patients; 6 months to 11 years of age	Bacteraemia	Turkey [188]
	Community acquired via central venous catheter	64-year-old male	Bacteraemia	USA [220]
	Hospital acquired	Healthy preterm neonate	Sepsis	Saudi Arabia [221]
	Hospital acquired via peritoneal dialysis	66-year-old male	Peritonitis	USA [222]
	Community acquired from plant and soil	42-year-old male	Peritonitis	Taiwan [73]
	Community acquired from plant and soil	87-year-old female	Septic shock	China [183]
** *Salmonella* ** **species**	Consumption of salmon	Multiple persons outbreak	Gastroenteritis	United Kingdom [223]
	Consumption of smoked halibut	Multiple persons outbreak	Febrile gastroenteritis	Germany [224]
** *Vibrio fluvialis* **	Shellfish	72-year-old male	Cholera-like diarrhoea	Mississippi USA [225]
	Following tidal wave	3,529 cases	Watery diarrhoea.	West Bengal eastern India [226]
	Contaminated freshwater and fish from these waters	Regular outbreaks during summer months	Acute enteric infections	Volga Russia [227]
	Swimming in contaminated water	Unknown	Waterborne otitis	Cuba [228]
	Swimming in contaminated water	40-year-old female	Waterborne otitis	Taiwan [229]
	Contaminated water or food	43 patients with diarrhoea	Diarrhoea	Kolkata, India [230]
	Brackish water	45-year-old male	Haemorrhagic cellulitis and cerebritis leading to amputation. Secondary to wound infection	Taiwan [231]
	Collecting fish in infected water with abrasion on leg	47-year-old male	Necrotising fasciitis and primary sepsis leading to amputation	Taiwan [232]
	Unknown	Outbreak of 10,674 patients	Diarrhoea	Dacca and Matlab Bazaar, Bangladesh [233]
	Cooked fish, raw oysters, shrimp	Twelve patients	Gastroenteritis, a wound in one patient, and a caecostomy drainage specimen in one patient	Florida, USA [234]
	Unknown	65-year-old male	Severe watery diarrhoea requiring parenteral hyperalimentation	Taiwan [235]
	Unknown	52-year-old female	Acute infectious peritonitis	Republic of Korea [236]
	Trivial cutaneous lesion or through gastrointestinal translocation after ingestion of undercooked seafood.	88-year-old female	Biliary tract infection	Taiwan [237]
	Two cultures from home tap water coming from well revealed heavy growth	52-year old female	Urinary tract infection	Beirut, Lebanon [238]

## Data Availability

Data are unavailable due to privacy restrictions.

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
