# Peer review of "Bacteria of Zoonotic Interest Identified on Edible Freshwater Fish Imported to Australia"

_foods, 2023, doi:10.3390/foods12061288_

Round 1
Reviewer 1 Report
A very interesting manuscript on the occurrence of bacteria of zoonotic interest on edible freshwater fish imported to Australia. The study is very well designed, the methodology is suitable for such a study, the results are clearly presented and adequately discussed. The outcome is a manuscript loaded with useful information. Only a few minor suggestions can be offered:
l. 59, the reference (Preena et al. 2020) has skipped numbering
l. 81, please define ‘FP’
l. 132-134, please explain more the calculations inside the parentheses, it can be really confusing
l. 263, why introduce c/g and what is the difference between c/g and cfu/g. Why not use the latter?
l. 417, word ‘Thesis’ seems to have escaped editing
Author Response
Reviewer # 1: A very interesting manuscript on the occurrence of bacteria of zoonotic interest on edible freshwater fish imported to Australia. The study is very well designed, the methodology is suitable for such a study, the results are clearly presented and adequately discussed. The outcome is a manuscript loaded with useful information.
All authors would like to thank Reviewer #1 for the encouraging feedback. Reviewer comments have been addressed in the revised MS.
- 59, the reference (Preena et al. 2020) has skipped numbering
Response: Thank you. Preena et al. 2020 is now numbered
- 81, please define ‘FP’
Response: Thank you this has now been defined as fishery products (FP)
- 132-134, please explain more the calculations inside the parentheses, it can be really confusing
Response: Authors apologise for the confusion. (Sp. A: six individual bags + 6 ‘bag water’ = 66 samples and Sp. B: (six individual bags + 6 ‘bag water’ = 47 samples). There were 6 individual bags of Sp. A (n = 60) + 6 bag water samples from each of the six individual bags of Sp. A which equals 66 samples in total. There were 6 individual bags of Sp. B (n = 41) + bag water samples from each of the six individual bags of Sp. B which equals 47 samples in total. The sentence has been revised to, ‘Bag water’ and Sp. A and Sp. B results were combined to reflect 66 and 47 samples, respectively, (Sp. A: six individual bags of fish (n = 60 fish) + 6 ‘bag water’ samples = 66 samples in total and Sp. B: (six individual bags of fish (n = 41 fish) + 6 ‘bag water’ samples = 47 samples in total)).
- 263, why introduce c/g and what is the difference between c/g and cfu/g. Why not use the latter?
Response: There is no difference it was just a preference of how the results were expressed. This has been changed throughout the MS according to reviewer comments
- 417, word ‘Thesis’ seems to have escaped editing
Response: Thank you. Thesis removed from caption
Reviewer 2 Report
Title: Bacteria of zoonotic interest identified on edible freshwater
fish imported to Australia
Manuscript No: foods-2262741
Reviewer Comments
The incidents of zoonotic disease in last few years. So research in this field is an good research area. Some shortcomings regarding some text. Below I have provided some remarks on the text.
In a research paper, So long Introduction is not required. 79 references are only in Introduction. So, only limited relevant information is required in Introduction.
Line 287: use standard units like hrs is a standard unit it is h. SO correct all units.
In Table 7 scientific names are not in italics.
Paper is well written, and very less mistakes are observed. English is also good.
I have suggested to include a paper in your discussion part which is about zoonotic disease (Yersinia) and how to control it: https://doi.org/10.3390/ijms232214090
Author Response
All authors would like to thank Reviewer # 2 for the constructive feedback
In a research paper, So, long Introduction is not required. 79 references are only in Introduction. So, only limited relevant information is required in Introduction.
Response: Thank you for the comment. Authors understand that the introduction is a little longer than average for a research paper. Examination of imported FP must be scientifically justified according to the trade agreements signed by Australia or the research may be deemed an unfair barrier to trade. In order to provide complete justification for the research conducted, authors were compelled to provide a slightly longer and rigorously researched literature review. Authors have reviewed the introduction to the MS and are satisfied that all information is required to support scientific justification for the examination of FP imported to Australia. The introduction is also slightly longer due to recommendations made by the PhD Examiner (M. Williams PhD candidate Thesis) who required certain inclusions in the Introduction.
Line 287: use standard units like hrs is a standard unit it is h. SO correct all units.
Response: Thank you. This has been corrected throughout the MS.
In Table 7 scientific names are not in italics.
Response: Table 7 and all other Tables in the MS have been checked to ensure all bacterial species are in italics.
I have suggested to include a paper in your discussion part which is about zoonotic disease (Yersinia) and how to control it:
Response: Thank you. Yersinia citation has been included, De Keukeleire, S.; De Bel, A.; Jansen, Y.; Janssens, M.; Wauters, G.; Piérard, D. Yersinia ruckeri, an unusual microorganism isolated from a human wound infection. New Microbes & New Infect. 2014, 2, 134-135).